# Metabolic Fingerprinting of Fabry Disease: Diagnostic and Prognostic Aspects

**DOI:** 10.3390/metabo12080703

**Published:** 2022-07-28

**Authors:** Maria Teresa Rocchetti, Federica Spadaccino, Valeria Catalano, Gianluigi Zaza, Giovanni Stallone, Daniela Fiocco, Giuseppe Stefano Netti, Elena Ranieri

**Affiliations:** 1Department of Clinical and Experimental Medicine, University of Foggia, 71122 Foggia, Italy; mariateresa.rocchetti@unifg.it (M.T.R.); daniela.fiocco@unifg.it (D.F.); 2Unit of Clinical Pathology, Center for Molecular Medicine, Department of Medical and Surgical Sciences, University of Foggia, 71122 Foggia, Italy; federica.spadaccino@unifg.it (F.S.); valeria.catalano@unifg.it (V.C.); elena.ranieri@unifg.it (E.R.); 3Unit of Nephology, Dialysis and Transplantation, Department of Medical and Surgical Sciences, University of Foggia, 71122 Foggia, Italy; gianluigi.zaza@unifg.it (G.Z.); giovanni.stallone@unifg.it (G.S.)

**Keywords:** Fabry disease, lysosomal storage diseases, metabolomics, LysoGb3, systems biology

## Abstract

Fabry disease (FD) is an X-linked lysosomal disease due to a deficiency in the activity of the lysosomal-galactosidase A (GalA), a key enzyme in the glycosphingolipid degradation pathway. FD is a complex disease with a poor genotype–phenotype correlation. In the early stages, FD could involve the peripheral nervous system (acroparesthesias and dysautonomia) and the ski (angiokeratoma), but later kidney, heart or central nervous system impairment may significantly decrease life expectancy. The advent of omics technologies offers the possibility of a global, integrated and systemic approach well-suited for the exploration of this complex disease. In this narrative review, we will focus on the main metabolomic studies, which have underscored the importance of detecting biomarkers for a diagnostic and prognostic purpose in FD. These investigations are potentially useful to explain the wide clinical, biochemical and molecular heterogeneity found in FD patients. Moreover, the quantitative mass spectrometry methods developed to evaluate concentrations of these biomarkers in urine and plasma will be described. Finally, the complex metabolic biomarker profile depicted in FD patients will be reported, which varies according to gender, types of mutations, and therapeutic treatment.

## 1. Introduction

Fabry disease (FD, OMIM #301500) is an X-linked inherited metabolic disease due to the deficiency of lysosomal-galactosidase A activity (GalA-EC 3.2.1.22), which is encoded by the GLA gene. This enzyme is crucial in the glycosphingolipid degradation pathway, thus leading to cellular dysfunction and microvascular pathology [1,2]. The incidence of FD ranges from 1 in 40,000 to 1 in 117,000 births in the general population [3]. However, this is believed to be widely underestimated as some screening studies reported higher incidence rates (e.g., 1/1500–1/7000) [4,5,6,7,8,9,10]. The impairment of GalA activity results in a progressive storage of glycosphingolipid derivatives such as globotriaosylceramide (Gb3) and galabiosylceramide in the lysosome (Figure 1). This may occur in several cell types such as endothelial, vascular, cardiac, renal and nerve cells, where the continuous deposition leads to serious cellular damage and organ failure [11]. Thus, damage to the heart, kidney and central nervous system will significantly worsen life expectancy [12].

Most Fabry patients show no symptoms in the first years of life; however, symptoms may arise in childhood or adolescence. The classical phenotype in FD is manifested by cornea verticillata, neuropathic pain, gastrointestinal dysfunction and angiokeratoma [13,14]. Serious complications usually occur in adulthood and may include progressive renal insufficiency, cardiac complications (arrhythmia, hypertrophic cardiomyopathy) and/or or nervous system involvement (central and peripheral) (Figure 2) [1,9,15,16].

Vascular ectasia and tortuosity could also be observed [13]. Pathogenic variants might present with low to absent residual GalA activity in males encompassing the full disease clinical spectrum, while in heterozygous females the presentation is miscellaneous due to the variable X-inactivation. More generally, the lack of GLA genotype–phenotype correlation may be due to putative modifier genes.

FD diagnosis is usually made by the deficiency in GalA activity in white blood cells from a blood sample, plasma/serum or a dried blood spot by using an enzymatic assay [17,18,19]. Other samples (i.e., lymphoblasts, cultured fibroblasts, tears or urine) could be used to identify a GalA activity deficiency [20]. Thereafter, the diagnosis is confirmed by molecular analysis of the GLA gene.

While reliable for male patients, the enzymatic activity assessment is questionable for female carriers given the abovementioned random inactivation of the X-chromosome. In this case, molecular analysis is mandatory to detect heterozygous individuals [21]. The storage of product Gb3 substrate is variously increased in either plasma [22,23] or urine [24] in late onset forms and in female patients. For a better discrimination, globotriaosylsphingosine (LysoGb3), a deacylated derivative of Gb3, is suggested. However false negatives in some female patients and in very late onset forms have been described [25,26]. Moreover, LysoGb3 does not correlate effectively with clinical events in patients under treatment, as recently illustrated in patients on chaperone therapy (migalastat) [27].

Regarding treatment, Enzyme Replacement Therapy (ERT) with intravenous exogeneous human-Galactosidase A has significantly enhanced FD management. To date, two ERTs are available: based either on recombinant Agalsidase [28] or on gene-activated human-Galactosidase A enzyme [29]. Both therapeutic options are able to improve the quality of life of the FD patients and to slow the course of the disease [30].

A valid management of FD requires an early diagnosis. However, the lack of robust surrogate markers and of molecular understanding of FD pathophysiology impairs or delays an effective diagnosis, patient stratification and personalized management [31,32]. Thus, a better understanding of FD biological plasticity might enforce our screening and diagnosis tools.

To this aim, the omics revolution has opened new perspectives to explore biological systems through data-rich strategies at an unprecedented width, depth and scope in different fields, including lysosomal storage diseases [33,34,35,36]. This omics research strategy is mainly driven by high throughput technologies, bioinformatics, data sciences and systems biology approaches. Such systems-based strategies promote unbiased, data-driven and hypothesis-free studies to explore health and disease states and get rid of hypothesis-driven aspects of conventional reductionist approaches [35]. In FD, several omics-based studies have been previously reported [37,38,39,40,41]. In this narrative review we report several metabolomics studies aiming to determine metabolic-based biological signatures that could discriminate Fabry patients from healthy subjects, and the different FD phenotypes among them. Moreover, we aimed to explore the clinical utility of metabolomics in FD diagnosis, prognosis and therapeutic monitoring. The bibliography search strategy we used was based on the following search keys: “lyso-gb3 AND mass spectrometry”, “lyso-globotriaosylsphingosine AND mass spectrometry”, “globotriaosylsphingosine AND mass spectrometry”, “fabry disease AND metabolomics”.

## 2. Metabolomics

Metabolomics is a high-throughput “omics” technology, which intends to give a picture of the whole biological system at the time it is observed in a hypothesis-free and unbiased mode [42]. This holistic investigative method, when applied to multiple levels of a given biological system, may offer the opportunity to deeply understand disease processes, allowing better diagnosis and treatment management. Indeed, description of biomolecules involved in a pathological process (genes, proteins, metabolites), the relations between them and their functions should be considered for a complete understanding of disease processes. Furthermore, bioinformatics and the development of computational advanced models allow the management of a huge amount of data generated by high-throughput omics technologies, by translating them into clinically effective tools to support medical decision-making in the new field of precision medicine [42,43,44].

The application of omics strategies to the study of biological systems revealed their complexity and provided insights for new approaches to disease diagnosis, prognosis and treatment. Metabolomics can measure low-molecular-weight metabolites associated with a disease phenotype, which may be useful to understand the metabolic pathway involved in the disease mechanisms, especially if complemented with proteome and genome data, whose changes are reflected in the metabolome [45,46].

In addition, only metabolomics offers the opportunity to analyze the dynamic variation of living organisms, irrespective of genotype. By its nature, metabolome changes faster than proteome, and this change speed makes it one of the most promising clinical tools for monitoring therapies or treatments [47]. In the following sections we will focus (i) on the main metabolomic studies which have underscored the importance of detecting biomarkers for a diagnostic and prognostic purpose in FD, this is also useful to explain the wide clinical, biochemical and molecular heterogeneity found in FD patients [1,48,49]; (ii) on quantitative mass spectrometry methods developed to evaluate concentrations of these biomarkers in urine and plasma; and (iii) on the complex metabolic biomarker profile encountered in FD patients, which varies according to gender, types of mutations, and therapies.

## 3. Metabolomics and Metabolic Diseases

If metabolome depicts the entire set of metabolites present in a biological specimen (urine, serum, cells, tissue, etc.), at a given time, metabolomics is the systemic biochemical characterization of the metabolites and their fluctuations related to genetic and environmental factors. Metabolomic technologies able to comprehensively define the biochemical profile of a given biological sample include mass spectrometry (MS) [50], nuclear magnetic resonance (NMR) [51], and Fourier Transform Near-Infrared Spectroscopy (FT-NIR) [52,53].

Although they should be used as complementary methods to provide a wider metabolome coverage, MS is one of the most used for both quantitative and qualitative analysis because of its high resolution, sensitivity and specificity. Gas chromatography (GC) [50,54] and liquid chromatography (LC) [50,55] are usually strategically coupled with MS to handle the complexity of biological specimens. Actually, high-resolution LC-MS-based metabolomics is the technique of choice for global metabolite profiling without bias, providing the simultaneous analysis of thousands of metabolites with a detection limit in the low nanomolar range and from minimal amounts of biological sample [50]. Further, it offers the possibility of targeted [56] and untargeted analyses allowing the discrete detection of a number of specific metabolites or the global metabolite profiling [57,58]. Untargeted metabolomics provides opportunities for new discoveries linking cellular pathways to biological mechanisms for a better understanding of cell biology and physiology [59,60]. Targeted metabolomics is often used for the investigation of a target pathway involved in a disease and perturbed during the different disease stages or by therapies. Commonly, this approach is driven by a specific biological question and requires the use of specific and quantitative analyses to provide meaningful and reliable responses [61].

Highly sensitive and robust methods to quantitatively and routinely measure numerous metabolites, with high throughput, have been developed using triple quadrupole (QqQ) MS to perform selected/multiple reaction monitoring (SRM/MRM) experiments [62,63]. MRM experiments are highly specific in metabolite identification and the possibility to use deuterated internal standards make this MS methodology an appropriate tool to quantitative measure metabolites changes [62,63]. Many targeted metabolomics analyses have been applied to investigate several types of inborn errors of metabolism, including FD [33,34,35,36,41,64,65,66,67,68,69,70].

## 4. Metabolomics in Fabry Disease

Metabolomic analysis is a very precious tool to understand at molecular level the pathological processes behind FD and to find molecules able to recognize and discriminate the numerous different disease phenotypes. The discovery of robust FD biomarkers may help lead to an early diagnosis and better patient stratification for an effective management of the disease and its therapeutic monitoring. Although the level of plasma and urine Gb3 has been found elevated in classic FD patients, its values vary in these patients, its normal plasma level in FD patients with N215S mutation and in heterozygotes did not help the characterization of FD females with clinical symptoms, normal α-GAL activity and N215S mutations [71]. Furthermore, Gb3 level does not seem to be adequate to monitor treatment unless it is elevated prior to ERT. On the other hand, urine Gb3 is generally elevated in FD females, thus it seems to be more useful than plasma Gb3 for the diagnosis of FD heterozygotes [71]. Along with that, the lack of correlation, in some FD cases, between Gb3 elevation and clinical manifestation, the discrepancy found between early storage of Gb3, in hemizygotes at or even before birth, and the onset of clinical symptoms, and, the lack of correlation of plasma or urinary levels of Gb3 with the disease severity in neither hemizygotes nor heterozygotes suggested the existence of another substance in addition to Gb3 that was involved in the disease pathogenesis. Indeed, the plasma level of globotriaosylsphingosine (LysoGb3), the deacylated form of Gb3, is increased in symptomatic FD patients (male and female) markedly exceeding that of Gb3 [25]. Because of (i) the inhibitory activity of LysoGb3 towards α-Gal A, (ii) its capacity to induce smooth muscle cell proliferation stimulating vascular remodeling, a characteristic feature of Fabry disease, and (iii) the reduction of its plasma levels by therapy, LysoGb3 was supposed to be useful to monitor Fabry disease [25,72].

In this context, the growing development of an increasingly sophisticated mass spectrometer capable of measuring molecules in biological fluids with improved sensitivity and accuracy has met the need to quantify LysoGb3, the only potential biochemical marker of FD. Auray-Blais and his group evaluated urine LysoGb3 as a biomarker in FD patients using time-of-flight (TOF) mass spectrometry and MRM analyses [73]. They (i) standardized an extraction method to make LysoGb3 detectable in urine, with high precision and sensibility (the limit of quantitation was 20 pmol/L–20 nmol/mL), by eliminating interfering compounds; (ii) confirmed the specificity of the methodology not finding LysoGb3 detectable in the urine of healthy controls; (iii) found correlations between urine LysoGb3/creatinine and Gb3/creatinine, gender, ERT status, and types of mutations, the last representing the parameter that most strongly influenced urine LysoGb3 concentrations. However, there was no correlation between urine LysoGb3 and estimated Glomerular Filtration Rate (eGFR), therefore urinary LysoGb3 does not seem a good indicator of renal function [73].

To confirm the importance of LysoGb3 as an FD marker, the need for its quantitative analysis in both urine and plasma samples arose. Gold and coworkers [74] accurately quantified plasma and urine LysoGb3, and the related lyso-ene-Gb3 (LysoGb3 containing one additional double bond) by using ultraperformance liquid chromatography–tandem mass spectrometry (UPLC-MS/MS) introducing an isotope-labeled LysoGb3 as the internal standard with the aim to overcome the detection limit (about 3 nmol/L) of the HPLC procedure normally used [72]. Quantification by MRM analysis revealed much higher LysoGb3 plasma concentration for male FD hemizygotes compared to FD heterozygotes and healthy controls, while lyso-ene-Gb3 concentrations were around 10–25% of total lyso-Gb3. On the contrary, Lyso-ene-Gb3 urine levels were comparable or even higher than LysoGb3, meaning that LysoGb3 could be poorly excreted in the urine, therefore reabsorbed by the kidneys. For this reason, the authors recommend the determination of LysoGb3 and lyso-ene-Gb3 in plasma for diagnostic purpose, also confirmed by other studies [75]. The method of Gold was the first which took advantage of an isotope-labeled internal standard with a chemical structure identical to the analyte, reaching a low detection limit for plasma LysoGb3 (0.05 pmol/mL) which had not been achieved by previous methods [76]. In fact, this method allowed (i) accurate quantification in both groups of FD hemizygotes and heterozygotes (males and females) although the wide linear dynamic range of LysoGb3 plasma concentration; and (ii) accurate quantification of plasma LysoGb3 in normal individuals, and in patients with an atypical clinical presentation of Fabry disease presenting only a slight plasma LysoGb3 increase [74].

At the same time, the group of Auray-Blais and Boutin, identified for the first time new LysoGb3 and related biomarkers in urine [77] and plasma [65,78] of FD patients by a metabolomic targeted approach. LC-TOF/MS metabolomic approach and multivariate statistical analysis identified higher levels of LysoGb3 and its seven analogues in FD male urine and plasma compared to females, while none of these biomarkers were detected in the majority of healthy controls. The LysoGb3 analogues which presented modified sphingosine moieties [65,77,78] (Table within the Figure 3) were excreted in higher levels in the urine of FD patients, as compared to the level of total LysoGb3. Moreover, some of them were reduced following ERT. Taken together, these data suggest that the LysoGb3 analogues measurement may be able to provide more information on disease status and response to treatment than Gb3/LysoGb3 alone. Subsequently, the same group standardized a multiplex tandem MS analysis for the quantification of the LysoGb3 analogues, in urine [66] and plasma [67] in a larger cohort of FD patients including pediatric Fabry patients [79]. The authors confirmed no detectable analogues in urine [66] and only traces of two LysoGb3 analogues (−2 and +34) in plasma [67] of some healthy controls, which makes LysoGb3 analogues good candidate disease biomarkers. Furthermore, higher LysoGb3 analogues urinary excretion levels in FD males compared to females were found and all of them were reduced after ERT in males [66]. They also observed that some specific LysoGb3 analogues (+O, *m*/*z* 802; +H_2_O_2_, *m*/*z* 820 and +H_2_O_3_, *m*/*z* 836) were more abundant in the urine of most patients, while lyso-Gb3 (+H_2_O, *m*/*z* 804) was not detected [66], anyway the evaluation of the global metabolic biomarker profile (Gb3, LysoGb3, and its analogues) has been suggested for FD diagnosis. In addition, for high-risk screening in FD patients of different ages it must be taken into account that FD children showed lower urinary LysoGb3 levels compared to adults [79]. In screening for FD diagnosis, the urinary excretion profile of LysoGb3 and its analogues may vary depending on the causal genetic alteration [79]. A discrepancy between the abundance of LysoGb3 and its analogues in urine and plasma was revealed: plasma LysoGb3 is more abundant than its related analogues [67], while it was the opposite in urine [66]. The metabolic relationship between plasma and urinary LysoGb3 analogues remains still unclear. This study [67] highlighted the need of high sensitive mass spectrometer in order detect in plasma some LysoGb3 analogues by the multiple reaction monitoring mode (MRM), in fact, LysoGb3 analogues (−28) and (+50) were not detected in a previous study [65]. The same authors published the protocol for the multiplex analysis of LysoGb3 and its six analogues [−C_2_H_4_ (−28), −H_2_ (−2), +O (+16), +H_2_O (+18), +H_2_O_2_ (+34), and +H_2_O_3_ (+50), Figure 3] in plasma [80] and a protocol for multiplex analysis of LysoGb3 and its seven analogues [−C_2_H_4_ (−28), −C_2_H_4_+O (−12), −H_2_ (−2), −H_2_ +O (+14), +O (+16), +H_2_O_2_ (+34), and +H_2_O_3_ (+50) Figure 4] in urine [81] based on a previously described method [66,67]. The last urine metabolomic signature was useful for the diagnosis of Fabry patients as well as for FD patients with cardiac variant mutations, and also to monitor the efficacy of ERT [81]. Interestingly, in a study including 12 FD patients carrying *GLA* gene variants, the assay of urinary and plasma metabolic profiles (Gb3, LysoGb3 and analogues) identified an increased level of urinary LysoGb3 analogue (+H_2_O_3_, +50) only in patients manifesting clinically severe heart disease, suggesting it might be an earlier biomarker of progressive heart involvement associated with FD [82].

A metabolomic approach was also used to study Gb3 isoforms in the urine [68] and plasma [41] of FD patients by using LC-TOF-MS/MS and multivariate analysis. By comparison between the urinary and plasma Gb3 analogues/isoforms, five novel Gb3 biomarkers were identified in plasma (including isomers at sphingosine and fatty acid moieties, and methylated form of Gb3 analogue) allowing one to discriminate FD patients from healthy controls [41] (Figure 4). This metabolic profile may be useful to understand the pathophysiology involved in FD, in addition it may represent a metabolic signature of FD patients after validation studies in a larger cohort of patients. Methylated Gb3 isoforms could be intermediary metabolites in the deacylation of Gb3 to generate the corresponding LysoGb3 [68], in fact, methylated intermediary metabolites were also found for galabiosylceramide (Ga2), another glycosphingolipid component of cellular membranes, found accumulated in vascular endothelium, nerves, organs and biological fluids of FD patients [69]. Ga2, along with its 22 isoforms, Gb3 and Gb3 analogues have been quantified in the urine of untreated male FD patients and healthy controls. Moreover, six Ga2-related isoforms have been hypothesized to be potential diagnostic biomarkers because they have been detected only in the FD group [69]; a hypothesis which is required to be validated in (the urine of) a larger cohort of FD patients comprising males/females, adult/children, and patients undergoing ERT treatment. Comparison between the abundances of Ga2 isoforms/analogs and their Gb3 counterparts revealed that they are involved in different anabolic/catabolic pathways. The same group published a protocol to detect a panel of 15 Gb3 analogs’ urinary biomarkers useful for screening, diagnosis, and long-term monitoring of FD patients [70]. This metabolite panel included methylated Gb3 isoforms which were particularly useful for screening late-onset cardiac variant mutations FD patients [70].

Recently, a network-based targeted metabolomic study discriminated FD patients from healthy individuals by a metabolic signature of 13 plasma metabolites including methionine sulfoxide, 2 lysophosphatidylcholines and 10 glycerophospholipids, underlining the important role of glycerophospholipids and oxidative stress in FD pathophysiology. Metabolites quantification was carried out using a commercial kit to extract plasma target metabolites analysis by LC-MS/MS using an isotopically labeled internal standard. A predictive model using the 13 metabolites showed an AUC-ROC of 0.992 in diagnosing FD patients. Moreover, the correlation of the metabolic signature with plasma LysoGb3 levels and GalA enzymatic activity highlighted the metabolic reshaping of FD, and the diagnostic potential of omics-based approaches in lysosomal diseases [64].

A metabolic reshaping in FD was also demonstrated by a glycosphingolipidomic study by LC-MS/MS on plasma and urine from a cohort of Fabry patients, grouped in symptomatic female, asymptomatic females, males and a healthy control [83]. Multivariate analysis showed 10 long chain isoforms of ceramide dihexosides C22:1, C22:0, C22:1-OH, C22:0-OH, C24:2, C24:0 C24:2-OH, C24:1-OH, C24:0-OH, C26:0 of Ga2 in urine are elevated in FD and were able to discriminate FD female patients better than Gb3 and lyso-Gb3, especially the asymptomatic female group (who are the most difficult to diagnose) from female controls [83]. These metabolites panels, after validation in a larger cohort of FD patients, could be combined together with LysoGb3/analogues and Ga2 for a potential powerful diagnostic tool, particularly for females where there is an uncertainty of diagnosis.

Numerous attempts have been made to discover Fabry biomarkers in urine as well as in plasma in the last decade by using LC-MS/MS analysis, which integrates excellent separation efficiency, high sensitivity, and specificity with analysis’ speed. A panel of urine and plasma Gb3 and LysoGb3 analogues/isoforms have been found with the high potential to discriminate FD patients from healthy controls. Further efforts must be made to validate these panels on larger cohorts of each FD phenotype, applying the protocols of multiplex LC-MS/MS analysis previously published or improving the existing one by using isotope labelled internal standards with increasing attention to the sample preparation [70,80,81].

## 5. Clinical Utility of GB3/LysoGb3 in FD Diagnosis and Prognosis

During the last decade, with the growing recognition of Gb3 and LysoGb3 as the most eligible candidate biomarkers of FD [25], the need for their measurement accuracy in biological fluids has been combined with the need to demonstrate their clinical usefulness. Therefore, researchers applied a more precise and accurate analysis protocol, using the robust LC-MS/MS method, for the measure of Gb3/LysoGb3 in a larger cohort of FD patients according to different phenotypes, age, gender, GLA mutations with the aim to assess their diagnostic and prognostic value. Most studies involved FD patients diagnosed by enzymatic (activity of the lysosomal GalA) or genetic (GLA gene analysis) assessment grouped based on gender, phenotype (classic vs later onset), age and the administration of therapies (ERT with recombinant human α-Gal).

A step forward in the assessment of the true utility of LysoGb3 for FD diagnosis and for predicting clinical outcome in FD patients has been undertaken by Talbot and colleagues [26]. They developed a simple LC-MS/MS assay for LysoGb3 measure that used only 10 uL of plasma without any extraction step as compared to Boutin’s [67,80] method which used 100 uL of plasma and solid phase metabolites extraction. By using MRM analysis, the authors identified:

(i) a cut-off of LysoGb3 level > 5 pmol/mL which was able to identify all male and more that 80% female FD patients with classical mutation (including five females with normal α-GalA);

(ii) no false positive in a cohort of 2000 samples from related non-FD lysosomal disorders (including Gaucher disease), which demonstrated high specificity of the plasma LysGb3 assay for FD diagnosis (<5 pmol/mL);

(iii) a drop in plasma LysoGb3 in classic mutation FD males only in the first 6 months following initiation of ERT, which stabilised thereafter, suggesting a response to therapy without future clinical outcome prediction.

However, LysoGb3 plasma assay did not identify FD patients with non-classical mutations, consistent with other papers [78,84], and it is not informative on clinical events although these were more prevalent in FD patients with higher LysoGb3 levels [26]. According to Talbot’s findings, Novak identified three GLA-mutations which proved FD females in whom the accumulating LysoGb3 was increased, whereas their leukocyte α-GalA activities were in the normal range [85]. These data supported the possible role of plasma LysoGb3 as a useful biomarker to improve the diagnosis of FD heterozygotes.

Although the utility of LysoGb3 as an FD diagnostic marker has been and is still debated [84,86], methods are constantly developing to make its measure easier by using, for example, whole blood rather than plasma. Ideally, biomarkers should be easy to collect, transport, and analyze, so the use of dried blood spots (DBS) started to be evaluated to measure LysoGb3. A high correlation between LysoGb3 concentrations in DBS and sera from 56 FD patients has been demonstrated. By highly sensitive LC-MS/MS, higher plasma LysoGb3 levels have been measured in affected males and heterozygotes with classic compared to later-onset phenotypes [23]. This study showed that LysoGb3 concentrations measured in DBS were almost 50% lower than those measured in sera, suggesting the development of more sensitive assays for LysoGb3. The same group, confirmed the association of serum LysoGb3 levels to phenotype severity, indeed it has been found higher in classic compared with later-onset in male and female patients and, for the first time, it has been found correlated with the genotype and disease severity [87]. Furthermore, higher LysoGb3 serum levels were found to be associated with serum creatinine and the presence of cardiomyopathy, and LysoGb3 serum levels were also higher among the males with severe mutations (deletions, insertions, duplications and some point mutations leading to a major change in the gene products) compared to males with missense mutations (individual point mutations that lead to single amino acid change) [87]. Plasma and urinary LysoGb3 and its analogues were also analyzed by LC-MS/MS in 34 FD patients with the late-onset N215S cardiac variant mutation, 62 classical FD patients and 109 healthy controls [88]. Plasma LysoGb3 levels > 2.7 nM differentiated N215S cardiac variant FD patients from no-FD individuals with diagnostic sensitivity and specificity of 100% [88]. Likewise, the analysis of urinary lyso-Gb3 and related analogues were able to diagnose all children patients bearing the p.N215S mutation, with normal urinary Gb3 levels, but elevated lyso-Gb3 analogues concentrations [79]. Once again, the importance of evaluating the entire metabolite profile of Fabry biomarkers for diagnosis has been demonstrated.

Additionally, the clinical effectiveness, including diagnostic value and disease surveillance, of plasma LysoGb3 in Chinese FD patients was reported using α-Gal A activity as a reference [89]. LysoGb3 levels were quantified by UPLC-MS/MS, and its diagnostic rate/power was similar to that of α-Gal A enzyme activity in male FD patients, while it resulted more useful at diagnosing female patients. A plasma LysoGb3 cut-off value of 0.81 ng/mL was able to separate patients with FD from healthy controls with 94.7% sensitivity and 100% specificity. Furthermore, in male FD patients serum LysoGb3 levels were more strongly correlated with disease severity than enzyme activity, instead no correlation was identified between LysoGb3 levels and disease severity in female patients likely due to the small number of female patients [89]. Later, the applicability of plasma LysoGb3 as a primary screening biomarker for classic and late-onset FD was confirmed by measuring LysoGb3 by UHPLC-MS/MS and plasma α-Gal A activity in male and female patients suspected of having FD based on clinical symptoms [90]. This prospective multicenter study, involving more than 2300 participants, established a plasma LysoGb3 cutoff value above 45 nmol/L (>2 ng/mL) to predict diagnosis of classic FD in males, while rather lower levels were found in late onset FD patients, requiring the integration of additional screening data (α-Gal A activity, and analysis of the *GLA* gene) to determine the boundaries between the two FD phenotypes. LysoGb3 screening had the potential to identify many unrecognized female probands (with normal α-Gal A activity), although LysoGb3 levels in asymptomatic females of families carrying class 1 mutations overlap with controls, as found in a previous paper [84], therefore, the problem to miss some female cases of FD, particularly late onset FD, still remained. To improve the diagnostic efficiency for FD in females, the α-Gal A/LysoGb3 ratio has been proposed as a novel biochemical criteria using dried blood spots (DBS) as test samples and HPLC-MS/MS analysis to measure LysoGb3. Interestingly, the cut-off value of α-Gal A/LysoGb3 ratio of 2.5 showed 100% sensitivity and specificity in distinguishing 35 female patients from 140 controls, whereas LysoGb3 levels were within normal range in 25.7% of females and the activity of α-Gal A presented normal in 91.4% of female patients with *GLA* gene pathogenic mutations and a family history of classic FD [91]. Noteworthy, the α-Gal A activity proved to be further higher than in other reports [92] and it may depend from study population and measurement methodologies. However, this issue deserves further discussion that is not within the scope of this review. Recently, a rapid simple and sensitive method for the analysis of plasma LysoGb3 by using the LC-MS/MS method and isotope labelled internal standard (LysoGb3-D7) identified a cut-off at 0.6 ng/mL of plasma LysoGb3. It allowed the discrimination of FD patients from healthy controls with 97.1% sensitivity, 100% specificity and 0.998 accuracy (ROC analysis), the diagnosis of patients with borderline α-GalA activity and individuals carrying non-pathogenic polymorphisms of the *GLA* gene [93]. It is noteworthy that no overlap was observed between the FD group (male and female) and the healthy controls or the nonpathogenic polymorphisms groups, allowing the assessment of atypical FD phenotype patients or classic FD females, who might show LysoGb3 plasma concentrations slightly above the experimental cut-off value due to residual αGalA activity [93].

Finally, a recent study focused on the usefulness of urinary Gb3 for diagnosis and monitoring FD patients with renal events [94]. A progressive, predominant accumulation of Gb3 in the kidneys gives rise to nephropathy onset in FD patients [1]. In this study, an easy and economic thin-layer chromatography-immunostaining method has been compared with the highly sensitive LC-MS/MS analysis. It revealed that urinary Gb3 level tended to be higher in the nephropathic FD compared to patients with no renal events in each phenotypic group of the disease, namely classic FD males, later-onset Fabry males, and Fabry females [94]. Since Gb3 originates from tubular cells and the urinary collecting system [95], and it has been found higher compared to liver and heart in both wild-type and Fabry mice [96], monitoring the development of nephropathy in Fabry patients exhibiting high levels of urinary Gb3 may be usefully suggested to understand its predictive power. We summarized the best diagnostic properties of different biomarkers in Table 1.

A wide range of plasma LysoGb3 cut-off values (0.6–8 ng/mL) [26,76,85,89,93] to classify FD patients from healthy controls have been published. They depend on many factors, essentially the heterogeneity of the sample size (often small), sex and age distribution (in both, patients and control groups), administration of therapy cause the reduction of case numbers in the individual groups to be compared, decreasing the performance of statistical tests. Then, differences in preanalytical (plasma sample collection and preparation) and analytical phase (internal standard, LC and mass spectrometer instruments used for biomarkers quantification) make it difficult to compare data between laboratories.

## 6. LysoGb3 and Its Analogues: Metabolic Signature to Monitor Therapy in FD

As reported elsewhere, the most widely used therapy for the clinical management of FD is ERT, while in selected cases an oral chaperone therapy with migalastat may be given [1,20]. Intravenous administration of ERT with both Algasidase alpha and beta increases the enzyme levels in the body, while oral pharmacological chaperones (PCT) have been shown to promote the correct folding of amenable mutated glycosidases and retrieve residual activity levels [20]. Plant-derived enzyme replacement therapy, substrate reduction therapy (SRT) and gene therapy using adeno-associated viral vectors are innovative therapeutic approaches currently under investigation in clinical trials. Beyond the main long-term clinical outcomes (reduction of cardiovascular events, renal insufficiency, neurologic involvement), it is useful to evaluate the impact of therapy onto the metabolic signature of FD, as it may predict the response to therapy and the clinical course of FD.

Regarding the impact of LysoGb3 levels and its analogues/isoforms on monitoring therapeutic treatment, most of the literature data indicated significant differences between the untreated and treated groups of FD patients in transversal studies and a significant decrease of LysoGb3 plasma concentration in follow up studies during treatment, especially in hemizygous patients. In patients with classical FD, different treatment regimens with agalsidase led to prominent reductions of plasma LysoGb3, at beginning (3–12 months) [97,98], thereafter LysoGb3 levels did not decrease further [97]. This phenomenon was much more consistent in males compared to females, because plasma LysoGb3 levels in heterozygotes were much lower [97]. The reaching of a plateau of plasma LysoGb3 levels has been attributed to the development of antibodies against α-Gal A in most of male patients, particularly those treated with agalsidase beta [97,98]. Antibody formation is revealed by lower decline of LysoGb3 in the first period of treatment; thus, LysoGb3 does not reach normal values [98] and it may reflect a worse treatment outcome. Indeed, monitoring the decline of the level of such biomarkers could be useful for clinical outcome prediction. The longitudinal multiplex analysis of plasma LysoGb3 and its analogues before and after ERT treatment in FD patients showed that their concentrations were lowered after the beginning of ERT, although they remained higher compared to their levels in a healthy control [67]. Interestingly, the lyso-Gb3 (+50) analogue was found to be completely absent after ERT, suggesting a possible indicator for monitoring treatment in these patients [67], which needs to be better comprehended in a larger number of treated male and female Fabry patients. A urinary panel of Gb3, LysoGb3 and related analogues was used to evaluate the follow-up of the treatment of male pediatric patients under ERT [79], evidencing differences in gender and age. First, ERT influenced the urine levels of all, Gb3, LysoGb3 and related analogues in FD male children lowering them after therapy, while in adult FD males only LysoGb3 (+34), lyso-Gb3 (+50) (Figure 1), and Gb3 showed a significant difference; in FD adult females ERT decreased urinary levels of LysoGb3, LysoGb3 (−28), LysoGb3 (−2) and Gb3 compared to untreated females. Although only two pediatric FD female patients followed ERT, reduction of urinary Gb3, LysoGb3, LysoGb3 (−28), LysoGb3 (−2) were observed after ERT compared to untreated Fabry females [79]. Since LysoGb3 (+50) analogue was reduced in both Fabry adult and pediatric males after ERT whereas it did not vary in treated versus untreated Fabry females, gender strongly influenced metabolic processes at cellular levels in the FD state.

## 7. Future Perspectives and Conclusions

FD is a rare disease, therefore randomized clinical trials (RCT) conducted on large cohort of patients are still lacking, and most literature data are represented by case series/prospective or retrospective studies on small numbers of patients. After reviewing all these data, we made the effort to summarize the biomarkers that until now have shown the best clinical performance (Table 1) with the awareness that high levels of evidence are absent. LysoGb3 may emblematically represent the metabolic phenotype of Fabry. Some issues arise in the LysoGb3 metabolic behavior for the diagnosis and prognosis of FD:-Plasma levels of LysoGb3 are significantly higher than in the healthy controls, more importantly they are higher in the Classic than in the Later-Onset phenotype in both male and female patients;-Plasma LysoGb3 levels are always higher in males than in females, however, higher LysoGb3 levels in family members of the same sex are higher in patients with elevated disease activity;-Within the Classic phenotype, plasma LysoGb3 levels are higher among the males with severe mutations;-Metabolic FD diagnosis is further complicated in females because most of them have normal or slightly decreased α-Gal A activities (indeed the α-Gal A activity assay is not reliable for females) with very low levels of plasma Lyso-Gb3.

It has been shown that a panel of metabolites in plasma (LysoGb3 analogues/isoforms [67], methionine sulfoxide, lysophosphatidylcholines and glycerophospholipids [64]) and in urine (Gb3 analogues26 and their methylated isoforms [70], Ga2 analogues/isoforms [69], ceramide dihexosides isoforms [83]) enables one to classify FD patients better than the single LysoGb3. This metabolites panel supports one of the dogmas of the omic approach in biomarker discovery, which is that given the complexity of a disease, more molecules together could better describe the molecular mechanisms involved in its pathological processes. However, all studies, which presented a metabolic panel for FD diagnosis or prognosis, needed to verify their findings in larger groups of patients. Furthermore, discrepancies were observed, especially in the efficacy of LysoGb3 analogues in differential diagnosis of FD patients and therapeutic monitoring, could be related to the small sample size and to the low detection limit of target molecules. This may be problematic for the discrimination of sample groups with very similar low levels of target molecules (i.e., later-onset FD patients, not classical FD patients, asymptomatic and/or treated FD female, healthy controls). After validation studies of the potential FD biomarkers, the analysis of the combined panel of metabolites in plasma or urine, by a robust LC-MS/MS-MRM method and the use of isotope labelled target molecules as am internal standard for accurate quantification, could offer the opportunity to better discriminate the different FD phenotypes. The use of the isotope labelled target molecules [74], instead of synthesized molecules similar to target molecules, favourably affect both their extraction efficiencies from biological fluid as well as the mass spectrometric behavior in different matrices. Lowering the detection limit of each biomarker, by optimizing the pre-analytical phase and mass spectrometry methods, could help to minimize overlapping among the group presenting lower biomarkers levels, therefore providing better patient stratification. Once a robust panel of metabolites are obtained as FD biomarkers, further efforts should be made to standardize the technical methods and inter-laboratory testing in order to compare measurements among laboratories. We hope that further future studies will reinforce the observations and the findings we report in our manuscript.

## Figures and Tables

**Figure 1 metabolites-12-00703-f001:**
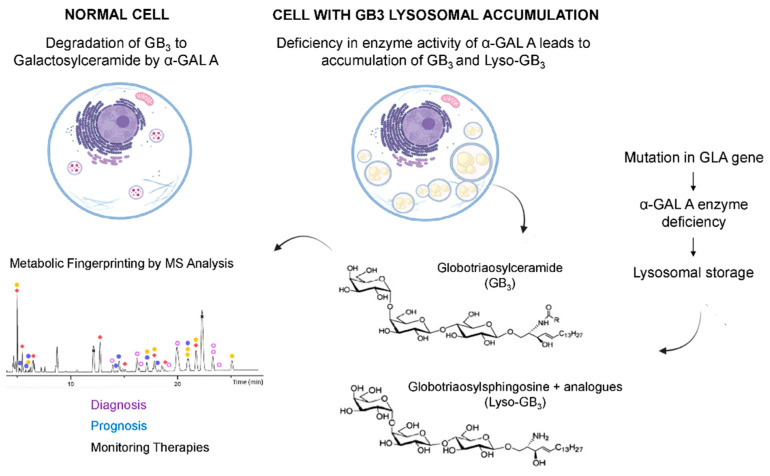
Pathophysiology of FABRY disease. The impairment of GalA activity results in a progressive storage of glycosphingolipid derivatives such as Gb3 or Lyso-Gb3 in the lysosome.

**Figure 2 metabolites-12-00703-f002:**
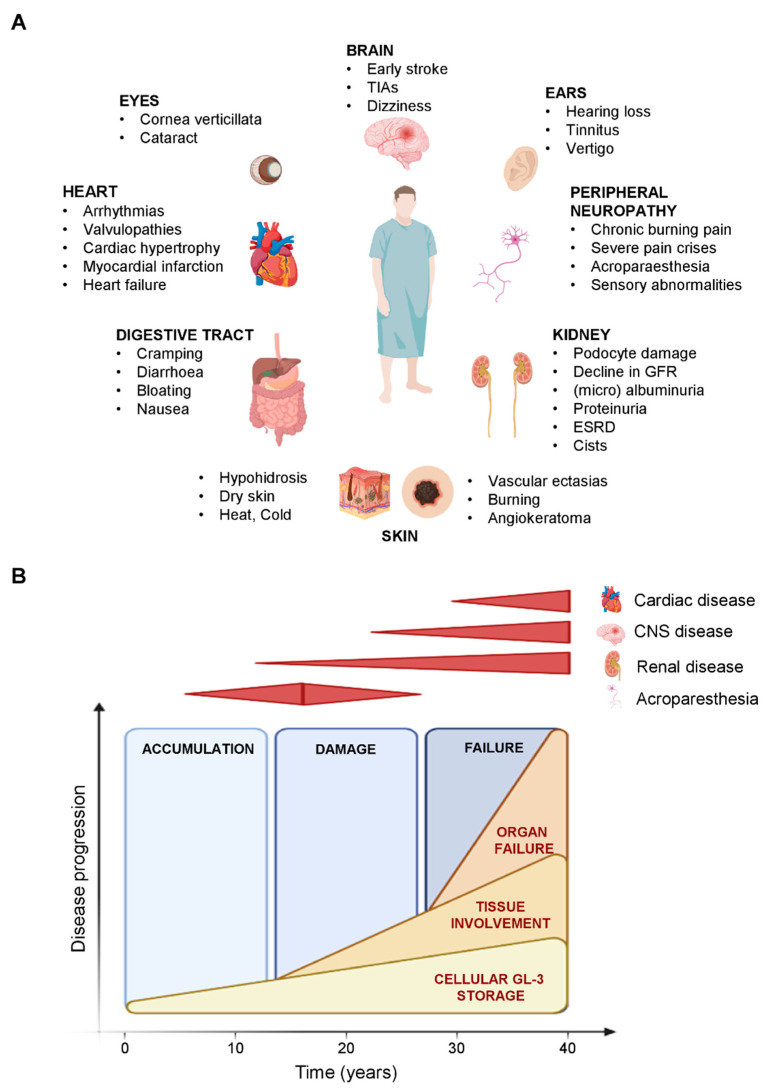
Clinical manifestations and disease progression in Fabry Disease. (**A**) Multisystemic organ involvement in Fabry Disease. (**B**) Model of disease progression related to major organs’ involvement in the classic form of Fabry Disease.

**Figure 3 metabolites-12-00703-f003:**
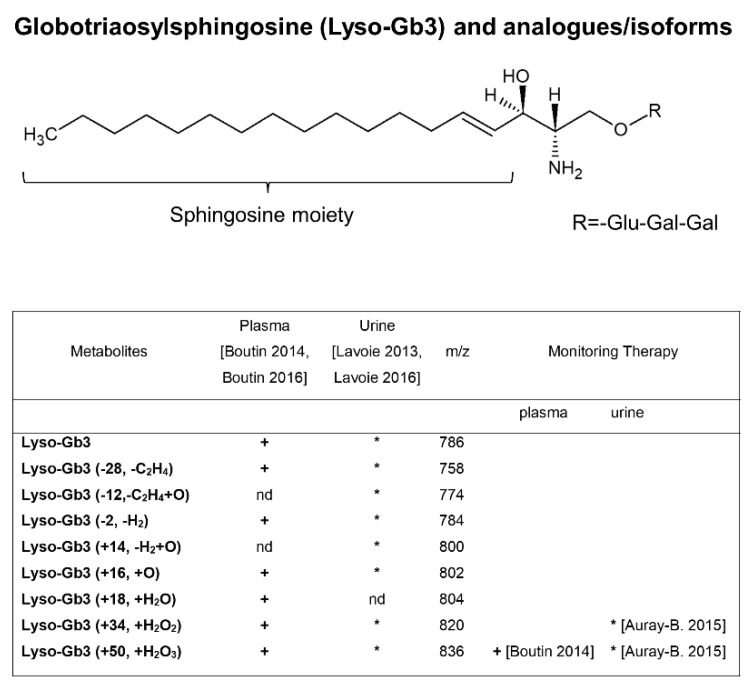
Structure of LysoGb3 and related analogues. LysoGb3 analogues have modification at sphingosine moieties (d18:1 = 18 carbon atoms with a single double bond). The table shows LysoGb3 analogues found in plasma [67,80] (+) and urine [66,81] (*) of Fabry patients as biomarkers for diagnosis/prognosis and monitoring therapy.

**Figure 4 metabolites-12-00703-f004:**
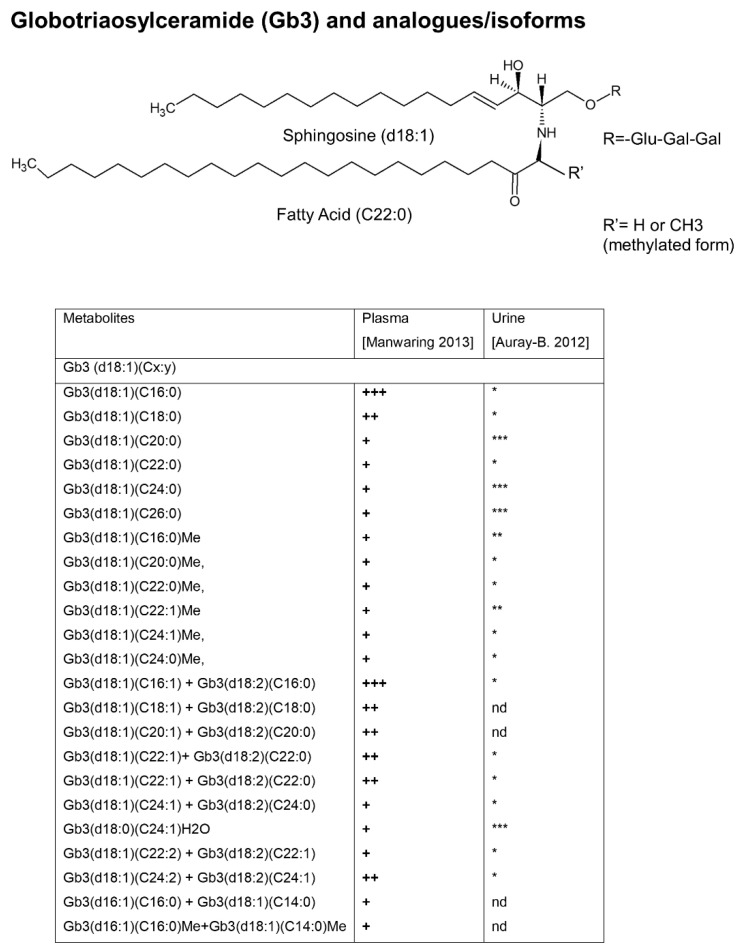
Structure of globotriaosylceramide (Gb3) and analogues/isoforms. Gb3 with behenic acid (C22), as native Gb3 and methylated Gb3. Glu: glucose; Gal: galactose. The table shows that Gb3 isoforms/analogues, detected in plasma of Fabry patients, are statistically different from healthy controls [41]. Some of Gb3 analogues/isoforms have been found in urine of Fabry patients [68]. Gb3 Isoforms: Gb3 with varying length fatty acid chains joined by an amide linkage to an unmodified sphingosine moiety. The main sphingosine species are described as d18:1 = 18 carbon atoms with a single double bond. Gb3 Analogues: Gb3 has modified sphingosine moieties. The structure of Gb3 isoforms and analogues is expressed in the following way Gb3(dv:w)(Cx:y)Z, where d indicates the sphingosine group, v shows the number of carbons in the sphingosine moiety, w shows the number of double bonds in the sphingosine moiety, C indicates the fatty acid group, x shows the number of carbon atoms in the fatty acid moiety, y shows the number of carbon−carbon double bonds in the fatty acid moiety, and Z indicates other modifications related to the sphingosine structure (e.g., methylated (Me) or hydrated (H_2_O)). +: detected in plasma; ++: abundantly detected in plasma; +++: highly abundant in plasma; *: detected in urine; **: abundantly detected in urine; ***: highly abundant in urine; nd: not detected.

**Table 1 metabolites-12-00703-t001:** Diagnostic properties of different biomarkers.

**Plasma**					
Biomarkers	Clinical Performance			FD population (n)	Reference
LysoGb3	AUC = 1 for each sex, with the best calculated cutoff for sensitivity and specificity at 34.8 ng/mL for males and 8.1 ng/mL for females to separate patients with FD from healthy individuals			Adult (69)	[87]
	AUC = 1, cutoff value of 2.7 nM yielded a diagnostic sensitivity and specificity of 100% for FD patients with the late-onset N215S cardiac variant mutation			Adult (96)	[88]
	AUC = 0.99, cutoff value of 0.81 ng/mL to separate male patients with FD from healthy individuals with 94.7% sensitivity and 100% specificity.			Adult (38)	[89]
	AUC = 0.99, cut-off value of 0.6 ng/mL between FD patients and healthy controls with 97.1% sensitivity and 100% specificity			Adult (34)	[93]
α-Galactosidase A/LysoGb3 ratio	AUC = 1, cut-off value of 2.5 with 100% specificity and 100% sensitivity to diagnose female FD from healthy individuals.			Adult (35)	[91]
**Urine**					
Biomarkers	Clinical Performance			FD population (n)	Reference
	Sensitivity % (Male-Female)	Specificity %	Accuracy %	Children (54)	[79]
Gb3	50–73	97	92		
LysoGb3	29–70	100			
LysoGb3 (−28)	54–87	100			
LysoGb3 (−12)	88–100	100			
LysoGb3 (−2)	50–83	100			
LysoGb3 (+14)	67–87	100			
LysoGb3 (+16)	88–100	100			
LysoGb3 (+34)	88–100	98			
LysoGb3 (+50)	42–91	92			

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
