# Peer review of "Metabolic Fingerprinting of Fabry Disease: Diagnostic and Prognostic Aspects"

_metabolites, 2022, doi:10.3390/metabo12080703_

Round 1

Reviewer 1 Report

Minor comments:

To change LysoGB3 to lysoGb3 (text and figures)

Angiokaratoma, acroparesthesia (in Figure 2)

Therapies in place of "therapeutic treatment (end of § 2.Metabolomics)

Vary in place of "variability" (§ 4 Metabolomics in FD)

Page 6:

The lysoGb3 analogues WHICH presented ... (the sentence is not clear)

The sentence after [ref 78] is too long

Page 8:

Multivariate analysis showed THAT ...

Figure 4

The table SHOW that ...

Reference 89 is debatable since about 40% of female patients with FD have normal alpha-galactosidase A activity (and not more that 90%). The authors have to discuss more about these results.

page 11

...originates FROM tubular cells...

page 12

I do not understand [nota]

... THESE patients|68].

page 13

the sentence beginning by However, all studies... is not clear, to rephrase

Author Response

We would like to thank the reviewer for the thoughtful comments and constructive suggestions, which help to significantly improve the quality of our manuscript.

We have made all corrections suggested by the reviewer in the text and in the figures, as highlighted by the red text in the revised version of the manuscript.

About the reference 89, in this case series the Authors describe that α-Gal A activity and lysoGb3 levels were altered in 8.6% and 74.4% of FD female patients, respectively. However, in the following sentence we underline that these data differ from what has already been reported in the literature (i.e. ref. 90) and suggest a possible explanation.

Reviewer 2 Report

This paper by Roccheti et al, reviews the metabolic fingerprinting of Fabry Disease for diagnostic and prognostic purposes considering the novel omics technologies. 

The whole manuscript is of interest since wide-validated and reliable biomarkers are lacking in Fabry Disease, especially in women.

The authors should receive credits for all their efforts. However, there are some minor concerns regarding the paper which should be addressed:

Minor Comments

Page 1. Abstract Section: The author stated that Fabry Disease could involve kidney, heart, or central nervous system. As the authors know, one of the features in classical form are acroparesthesias and dysautonomia due to peripheral nervous system involvement. 

Page 2. Introduction section, line 1.  Please peripheral nervous system involvement might be mentioned. For example, cardiac complications (arrhythmia, cardiomyopathy) and/or nervous system (central and peripheral).

I recommend a table that summarizes the diagnostic properties (e.g sensitivity, specificity, and accuracy) of different biomarkers reviewed in the different clinical scenarios. This table would improve the comprehension of the paper.

Author Response

We would like to thank the reviewer for the thoughtful comments, which help to significantly improve our manuscript.

We emended the abstract section and the introduction, with the mention of the peripheral nervous system involvement (red text). Finally, we added a table to summarize the best diagnostic properties of different biomarkers, as reported in the medical literature.

Reviewer 3 Report

This review manuscript is comprehensive, well-written and of interest to those caring for patients with Fabry disease. However, the authors prepared their manuscript in a classic "narrative" style which do not comply with modern requirements of high-rank journals. The authors need to reveal how they search and select the available literature best shown as a diagram. They also need to show strengths and limitations of their search methods and more critically present the results. Figure 2 seems to be redundant since it provides no new information and its content may be easily contained in the main text

Author Response

We would like to thank the reviewer for his thoughtful comments and constructive suggestions, which prompted us to re-evaluate our manuscript overall to improve it significantly.

With regard to the  request to specify the bibliography search method, we agree with the reviewer on the need to use a systematic approach as much as possible. About  our work, however, we chose to write a narrative review (as stated in the revised Abstract) as no randomized clinical trials (RCT) conducted on large cohort of patients were available. This deficiency is soon explained by the fact that FD is a rare disease and often the manuscripts are case series / prospective or retrospective studies on small numbers of patients.

For this reason, we have reported all the main manuscripts that met bibliographic search criteria, using the following search keys: lyso-gb3 AND mass spectrometry”, “lyso-globotriaosylsphingosine AND mass spectrometry”, “globotriaosylsphingosine AND mass spectrometry”, “fabry disease AND metabolomics”.

The cited manuscript reported the updated knowledge on the diagnostic and prognostic role of metabolomics biomarkers for FD, despite the absence of high levels of evidence. We hope that further future studies will reinforce the observations we report in our manuscript.

Regarding Figure 2, we agree with the reviewer that it does not provide further information as compared to the main text. However, we believe that, in the context of a narrative review, it can facilitate the knowledge of the disease for a Journal’s reader who approaches for the first time the topic. Therefore we propose to keep it in the final version of the manuscript.

Round 2

Reviewer 3 Report

The authors slightly improved the manuscript however still some important elements of a review are missing and need to be included. They include the search strategy the authors used and listing of the weaknesses and strengths of their work. The figure 2 may remain but in its lower part it needs to be added that it only applies to the classic form of FD and years are missing on an X axis. 

Author Response

According to the reviewer's suggestion, we have included the strategy search and the weaknesses and strengths in the revised version of the manuscript (red text). Moreover, we have improved Figure 2 (lower part) and its caption.

We would like to thank the reviewer for the constructive suggestions, which help to improve the quality of our manuscript significantly.